# Effects of Off-Season Heavy-Load Resistance Training on Lower Limb Mechanical Muscle Function and Physical Performance in Elite Female Team Handball Players

**DOI:** 10.3390/jfmk9040268

**Published:** 2024-12-12

**Authors:** Bjørn Fristrup, Peter Krustrup, Anders Kløve Petz, Jesper Bencke, Mette K. Zebis, Per Aagaard

**Affiliations:** 1Research Unit for Muscle Physiology and Biomechanics (MoB), Department of Sports Science and Clinical Biomechanics, University of Southern Denmark, 5230 Odense, Denmark; akpetz@health.sdu.dk (A.K.P.); paagaard@health.sdu.dk (P.A.); 2Department of Sports Science and Clinical Biomechanics, SDU Sport and Health Sciences Cluster (SHSC), University of Southern Denmark, 5230 Odense, Denmark; pkrustrup@health.sdu.dk; 3Danish Institute for Advanced Study (DIAS), University of Southern Denmark, 5230 Odense, Denmark; 4Sport and Health Sciences, University of Exeter, Exeter EX1 2LU, UK; 5Human Movement Analysis Laboratory, Department of Orthopedic Surgery, Copenhagen University Hospital, Amager-Hvidovre, 2650 Hvidovre, Denmark; jesper.bencke@regionh.dk; 6Department of Midwifery, Physiotherapy, Occupational Therapy and Psychomotor Therapy, University College Copenhagen, 2200 Copenhagen, Denmark; mzeb@kp.dk; 7Institute of Sports Medicine Copenhagen, Copenhagen University Hospital-Bispebjerg and Frederiksberg, 2400 Copenhagen, Denmark

**Keywords:** resistance training, agility performance, jump capacity, knee extensor and flexor strength, explosive muscle strength, sprint ability

## Abstract

**Background/Objectives**: Team handball involves a high number of rapid and forceful muscle actions. Progressive heavy-load resistance training is known to enhance mechanical muscle function; however, its transfer into functional performance in team handball athletes remains largely unknown. The current study aimed to analyze the effects of eight weeks undulating heavy-load resistance training on lower limb mechanical muscle function and sports-specific performance in elite female team handball players. **Methods**: Players from the Danish Women’s Handball League were block randomized to perform an off-season resistance training program (RT, *n* = 12, 23.0 ± 2.7 yr) or follow a training-as-usual control program (CON, *n* = 15, 24.1 ± 3.8 yr). All study participants were tested before and after an eight-week period during the off-season phase, including assessments of maximal isometric knee extensor and flexor peak torque, rate of torque development, countermovement jump (CMJ) power/work, and sports-specific performance (maximal vertical countermovement jump height, sprint capacity, team handball-specific on-court agility). **Results**: Agility performance improved for RT (−3.5%, *p* = 0.008), different from CON (*p* < 0.001) following eight weeks of designated resistance training. Additionally, CON demonstrated impaired agility (+4.0–7.3%, *p* < 0.05) and 20-m sprint (+1.9%, *p* = 0.002) performance. Maximal knee extensor peak torque increased in RT (4.5%, *p* = 0.044). Vertical CMJ flight height (JH) increased in both groups (RT +4.8%, *p* = 0.012, CON +8.4% *p* = 0.044); however, jump height relative to ground level (JHGL) increased in RT only (+8.0%, *p* = 0.013). **Conclusions**: In conclusion, designated resistance training during the off-season period is effective in maintaining and improving essential components of sports-specific performance and maximal knee extensor strength in elite female team handball players. Comparable protocols of twice-a-week heavy-load resistance training may also be beneficial in other types of intermittent elite team sports (i.e., football, basketball) that include maximal jumping actions, short-distance sprints, and rapid change of direction movements.

## 1. Introduction

Elite female team handball players encounter the challenge of juggling technical/tactical team handball training, weekly competitive matches, in-season strength and conditioning sessions, and personal life [1]. This leaves minimal time to focus on specific training modalities for optimizing sports-specific performance (i.e., vertical jumping, sprint capacity, agility) and even less time to participate in research activities involving extensive periods of lab-based testing. However, during the off-season, when there is a reduced emphasis on technical/tactical training as well as competitive matches, a potential opportunity arises for optimizing physiological performance and conducting on-site performance testing.

Elite female team handball is a physically demanding intermittent sport with a high proportion of aerobic energy expenditure, interspaced by bouts of anaerobic intense actions such as jumps, turns, sidecuts, accelerations, and decelerations [2,3]. Therefore, the ability to perform rapid and forceful muscle contractions represents a physiological advantage in team handball gameplay. An assessment of the explosive contractile properties of selected lower limb muscles may be performed by analyzing the characteristics of the torque–time curve during isometric muscle force production (rate of force/torque development—RFD/RTD) [4,5].

It is well established that heavy (>70% of 1 RM), progressive resistance training programs can be used to improve maximal voluntary isometric contraction (MVIC) peak torque, while concurrently evoking increases in the rate of torque development (RTD) in untrained as well as trained individuals [6,7,8,9,10,11,12]. To improve maximal muscle strength in trained individuals, it is recommended to use undulating resistance training programs [13]. Additionally, improvements in muscle strength and power are observed in male team handball players after eight weeks of heavy-load resistance training, with two sessions per week [14]. However, the plasticity in maximal muscle strength and RTD in response to heavy-load resistance training is less clear in female athletes [3,6].

Multi-joint countermovement jump (CMJ) testing has been used to assess the eccentric–concentric stretch-shortening cycle (SSC) power and RFD abilities of the lower limb extensor muscles [15,16], which may have strong ecological validity due to a high transferability to given locomotion activities of team handball.

The sports-specific performance of elite team handball players has been evaluated by various jump, sprint, and game-specific performance tests, including timed agility/change of direction tests [17,18,19,20,21,22,23]. In match play, team handball players frequently engage in high-intensity short-distance sprints (≤10-m), as well as sprints at near-to-maximum speed [3,24,25], making sprint capacity an important quality. A novel team handball-specific agility test has been developed by Team Danmark (the Danish Elite Sports Institution) in close collaboration with DanskHåndbold (the Danish Handball Federation). The agility test comprises handball-specific movements, short sprints, horizontal jumping, decision making, sidecutting, sidestepping, and horizontal changes of direction, aiming at evaluating movement performance. Notably, however, to our best knowledge, no previous study have examined the effect of lower limb heavy-load resistance training on this particular agility test with elite female team handball players. The reliability of field-based test methods for the assessment of mechanical muscle function and sports-specific performance has previously been reported by Fristrup et al. (2024), demonstrating the high test–retest reliability of CMJ testing on a force plate in kinetic (power, work) and kinematic (jump capacity, body center of mass displacement - BCM_disp_) parameters. Additionally, the reliability of isometric knee extensor and flexor peak MVIC torque, measured in a portable isometric dynamometer, were excellent. Furthermore, Fristrup et al. (2024) assessed the test–retest reliability of a 20-m sprint, measured by photocells, and found good-to-excellent reliability in 5-m, 10-m, and 20-m performance.

The aim of the present study was to employ on-site tests to evaluate the effects of an eight-week, undulating, progressive heavy-load resistance training program during the off-season phase in elite female team handball players. The specific focus was to evaluate selected outcomes of lower limb mechanical muscle function (MVIC strength, RTD, power) and sports-specific physical performance (jump and sprint capacity, agility).

## 2. Materials and Methods

### 2.1. Study Design

The present study was designed as a randomized controlled trial in elite female team handball players. Study participants were randomized to either perform eight weeks of designated undulating heavy-load resistance training (RT group) or to follow a training-as-usual program (CON group) during the off-season period from late May to late July 2023. Stratified block randomization was used to allocate players to either RT or CON. Players were stratified into four blocks, so that one block consisted of the all the players from the same team handball club. Each block/club was given a number from 1 to 4. A simple randomization was then performed to assign each block/club to one of the two groups, RT or CON, using www.random.org to generate a random sequence of four numbers (1–4). In advance, it was decided that the first two numbers in the sequence were to be allocated to RT and the last two numbers to CON.

Study participants were tested before (PRE) and after (POST) the eight-week intervention period. PRE-testing was completed within one week from the beginning of the intervention period, while POST-testing was completed one week after the end of the intervention period. Prior to PRE-testing, all subjects were familiarized with the testing protocol in separate test sessions (a maximum eight weeks prior to PRE-testing). Study participants were instructed to refrain from strenuous physical activity 24 h prior to the PRE- and POST-test sessions and to abstain from caffein intake on the days of testing.

### 2.2. Ethics and Delarations

This randomized controlled trial was registered at the Regional Committees on Health Research Ethics for Southern Denmark (20212000-114) and at ClinicalTrials.gov (ID: NCT06356935). The study was funded by Team Danmark through the Novo Nordisk Foundation grant to Team Danmark (Grant number NNF.22SA0078293), as one of the PRoKIT research network studies (Performance, Recovery, and Diet Optimization in Intermittent Sports). Part of the study was conducted in the Team Danmark National Elitesport Center, Brondby, Denmark, funded by the grant to Team Danmark from the Novo Nordisk Foundation (grant number NNF.22SA0078293). The authors declare no conflicts of interest.

### 2.3. Subjects

Elite team handball players from four clubs in the Danish Women’s Handball League were recruited for this study. After written and oral information, 60 elite female team handball players agreed to participate in this study. Participants were excluded if they had been absent from team handball training for more than two weeks due to injuries within the six months preceding PRE-testing or if they sustained injuries during the intervention period. Additionally study participants were excluded from the study if they became pregnant or ceased to play team handball at elite level. A total of 33 players withdraw from the study during the intervention period due to career stops, club changes, injuries, pregnancies, logistical obstructions on test days, or personal reasons (Figure 1). Twenty-seven players (RT *n* = 12, CON *n* = 15) were assessed in this study.

### 2.4. Instruments and Procedures

All field-based testing was conducted in the participating clubs’ respective facilities or at other nearby local facilities. The test protocol consisted of questionaries, body composition measurements, warm-up procedures, a countermovement jump test, tests of knee extensor and flexor MVIC strength and RTD, and a test of sports-specific performance in sprint and agility (Figure 2). The reliability for the investigated test methods has previously been described to be good-to-excellent (ICC ≥ 0.70, CV_w-s_ ≤ 10%) for countermovement jump testing, knee extensor and flexor strength, and 20-m sprint testing, respectively [4].

#### 2.4.1. Questionaries

Study participants filled out an online questionnaire about their playing position, current and previous team handball experience, and experience with resistance training (see Table 1).

#### 2.4.2. Body Composition

Bioimpedance (Inbody 270, InBody Co., Ltd., Seoul, Republic of Korea) was used to assess body composition in terms of body mass (BM), skeletal muscle mass (SMM), and fat percentage (FAT%). The average of two height measurements (Leicester Height Measure Mk II, Child Growth Foundation, Newcastle, UK) was reported (see Table 1).

#### 2.4.3. Warm-Up Procedures

Prior to the testing of maximal effort, all study participants completed a standardized ~10 min warm-up program. The program was previously described in Fristrup et al. (2024); in brief, it included a variety of exercises, such as low-to-high intensity runs, body-weight strength exercises, change of direction movements, and ball throws. Notably, all exercises incorporated the use of a handball.

#### 2.4.4. Countermovement Jump Testing

Stretch-shortening cycle (SSC) muscle performance was evaluated by means of bilateral CMJ testing on an instrumented force plate (AccuPower, AMTI, Watertown, MA, USA). Subjects performed two submaximal warm-up jumps, succeeded by five maximal single-effort jumps, separated by 30 s recovery between each jump. Study participants were instructed to perform the jump in a continuous motion, focusing on jumping as high and rapidly as possible with their hands on theit hips [4]. Vertical ground reaction force (F_z_) signals were A/D converted (16-bit) and sampled at 1000 Hz using a custom-built software script (MATLAB (R2024b), Mathworks, Natrick, MA, USA). The F_z_ signals were subsequently exported and examined using another custom-built software script (MATLAB, Mathworks, Natrick, MA, USA). The jump displaying the greatest jump height (JH) was selected for further statistical analysis. Maximal vertical jump height (JH) was calculated as $V_{\mathrm{to}}^{2}/2\;g$ where vertical velocity at toe-off (V_to_) = ∫(F_z_/BM − g)d*t*, BM = body mass, and g = 9.81 m/s^−2^ [4,26]. Jumps were divided into the initial eccentric phase [Ep] (body center of mass (BCM) moving downward), followed by the concentric phase [Cp] (BCM moving upward) [27]. Specifically, Ep was defined as the time interval for downward BCM movement to reach its deepest position (velocity (V) = 0), which was divided into an eccentric acceleration phase [Ep_acc_] representing the time interval from the onset of downward movement to the instant of maximal negative (downwards) BCM velocity (V_peak_ [E_p_]), continuing into the eccentric deceleration phase [Ep_dec_], representing the time interval from V_peak_ [Ep] to the deepest BMC position, which continued into the concentric movement phase [Cp], representing the time interval with an upward BCM movement from its deepest position (V = 0) to take-off (feet leaving the force plate). In addition to JH, jump performance was analyzed for peak power exerted on the BCM, rate of force development (RFD) in the Ep_dec_ (0–50 ms), concentric work [Cp], and the magnitude of BCM displacement in the eccentric (BCM_disp_ [Ep]) and concentric (BCM_disp_ [Cp]) phases, respectively. In addition, the time (T) of Ep_dec_ and Cp was recorded. The peak force of the eccentric and concentric phase (peak F_z_ [Ep], peak F_z_ [Cp]) was registered, along with a calculation of mean F_z_ [Cp]. Finally, the jump height relative to ground level (JH_GL_) was calculated as JH + (BCM_disp_ [Cp] − BCM_disp_ [Ep]) [26]. Peak power, RFD, work, peak, and mean F_z_ were normalized to BM.

#### 2.4.5. Isometric Knee Extensor and Flexor Strength and RTD

The maximal isometric voluntary strength (MVIC) was obtained for the knee extensors and flexors in the dominant leg (take-off leg during jump-shooting), along with an assessment of RTD during the phase of rising muscle force (0–100 ms relative to force onset) using a portable isometric dynamometer (Dynamometer, Science to Practice (S2P), Ljubljana, Slovenia). The anatomical knee joint angle was measured with a goniometer (Baseline^®^, HiRes^®^ 360° 30 cm, Fabrication Enterprises Inc., White Plains, NY, USA). Knee extensor and flexor MVIC torque were obtained at anatomical knee joint angles of 66° (RT: 66.4° ± 3.2°, CON: 65.5° ± 2.9°) and 45° (RT: 44.5° ± 5.1°, CON: 43.3° ± 3.8°), with the lever arm = 0° being horizontal. The seat position and lever arm were individually adjusted for each study participant, with identical positionings at PRE and POST testing, ensuring the alignment of the rotational axis of the lever arm with the medial femoral epicondyle. Additionally, the ankle cuff was individually adjusted, being positioned approximately 2 cm above the lateral malleolus [4]. Participants were firmly strapped to the rigid chair at the hip and distal thigh [28]. Individual seat and ankle cuff positions were the same at all three test sessions.

Study participants performed two submaximal contractions, succeeded by five maximal voluntary effort contractions for knee extensors, followed by five maximal voluntary effort contractions for the knee flexors, with a one-minute pause between each trial [4,29]. Participants were carefully instructed to contract as rapidly and forcefully as possible, maintaining the contraction for 4–5 s or until on-screen visual torque decreased. The researcher offered verbal encouragement, and participants received online visual feedback on a PC screen [30]. Knee joint torque was measured using a strain gauge-based torque sensor (model Z6FC3-200 kg, Hottinger-Baldwin Messtechnik GmbH, Darmstadt, Germany). The strain gauge signals were A/D-converted and sampled at 1000 Hz (ARS dynamometry, S2P Ltd., Ljubljana, Slovenia). All torque recordings were corrected for the effect of gravity on the lower leg [4]. Raw torque signals were exported for subsequent data analysis in a custom-build analyzing software script (MATLAB, Mathworks, Natrick, MA, USA). To remove any high-frequency noise, all torque signals underwent lowpass filtering using a digital fourth-order, zero-lag Butterworth lowpass filter with a cutoff frequency of 15 Hz [29]. The threshold to identify the onset of torque was defined as baseline torque +1% of MVIC peak torque. Attempts with pre-contracting (baseline) torque exceeding 1% of MVIC were discharged. For further statistical analysis of knee extensor and flexor peak torque, the trial with the highest MVIC peak torque was chosen. The rate of torque development was determined as the slope of the torque–time curve (Δtorque/Δtime) and calculated in an early-phase time interval, 0–30 ms, and late-phase time interval, 0–100 ms (T = 0 denoting onset of contraction) [4,31]. Impulse was calculated as the area under the torque–time curve (∫Torque d*t*) for the aforementioned early- and late-phase time intervals [4] (data not reported). Impulse reflects the angular momentum (and hence speed of the lower limb if it had been allowed to move freely); thus, impulse may be considered the most functional measure of rapid muscle torque production. For the assessment of RTD values, the trial with the highest impulse from 0 to 200 ms was selected [4]. Kinetic variables were normalized to BW.

#### 2.4.6. Sprint Performance

Using single-beam photocells (8 MHz Wireless Training Timer (WITTY-gates), Microgate, Bolzano, Italy), positioned at a height of 75-cm above the ground, sprint performance was assessed in a 20-m maximal sprint test, with the target finish line set at 25-m and visual markers at 5-m, 10-m, 15-m, 20-m, and 25-m. Sprint split times were recorded at 5-m, 10-m, and 20-m. Study participants initiated the sprint from a standing start position, with the front foot positioned behind the start line and the start sensor placed 20-cm behind the start line on the floor, causing the sensor beam to be interrupted by the malleolus of the front foot. The time started when the front foot was lifted from the starting position. Before performing five maximal sprints interspaced by a 1 min pause, participants completed two submaximal warm-up sprints at approximately 60% and 80% of maximal intensity, respectively. Subjects started on their maximal sprints at own initiative whenever ready, following the 1 min pause and within a 15 s window. The fastest 20-m sprint time was chosen for subsequent statistical analyses of 5-m, 10-m, and 20-m split times.

#### 2.4.7. Agility Performance

Agility performance was evaluated using a specific team handball game-based test developed by Team Danmark (the Danish Elite Sports Institution) in close collaboration with DanskHåndbold (the Danish Handball Federation) (Figure 2). In brief, motion and impact sensors (FITLIGHT Trainer^TM^, FITLIGHT Sports Corp., Aurora, ON, Canada) were used to evaluate agility performance. Participants initiated the test by removing their hand from the start sensor positioned 1-m behind the start line. Subsequently, participants ran for 3 m, before jumping over a 20-cm high hurdle and landing within a 100-cm × 50-cm box with toes oriented forward. While in the air, a directional light, either to the right or left, was displayed for 0.5 s. Following the landing, participants made a sidecut to either the right or left and touched a sensor at a height of 35-cm positioned on the 6-m line. Next, subjects sidestepped for 4-m and touched another sensor at a height of 1.5-m, before concluding the test with a 5-m straight-line sprint to the finish line. First, study participants completed six trials without a handball (see Figure 3a); subsequently subjects completed six trials with handball dribbles (see Figure 3b). Each set of six trials was randomized to either the left or right side. A random sequence of 6 numbers (1–6) was obtained using www.random.org, with unequal numbers being the left and equal numbers being the right side. In trials with handball dribbles, subjects were instructed to take two steps before executing the first dribble, followed by executing one dribble after the landing but before touching the sensor (during the sidecut movement), then executing two dribbles while sidestepping (changing the hand used for dribbles), and executing two dribbles while sprinting towards the finish line (see Figure 3b). The best total time to complete the agility test was chosen for subsequent analysis.

### 2.5. Resistance Training Intervention

Study participants in the RT group were offered two supervised training sessions per week for eight weeks. RT participants were encouraged to attend the supervised training sessions at a local fitness center with restricted public access. The designated resistance training program was designed to enhance mechanical muscle function and to improve the sports-specific performance of the lower limbs. Using undulating periodization on a weekly basis [13], two sessions were performed each week, with weekly Session #1 focusing on explosive muscle strength development using heavy-load exercises performed as fast as possible during the concentric contraction phases of the lifts and weekly Session #2 focusing on increasing the tolerance to high metabolic stress, using heavy-load slow movements. The program was progressively adjusted from 3–4 sets with 10–12 repetitions in the first weeks, to reach 3–4 sets with 4–6 repetitions in the last weeks [10] (Table 2). All loads were >70% of 1 RM (heavy), gradually increasing from 70% of 1 RM (12 reps) to 92% of 1 RM (4 reps) with decreasing repetitions [10,32]. The rest time between exercises was 1–2 min, rest time between sets was 2–3 min, and restitution time between sessions was ≥48 h. Each session consisted of three primary exercises combined with secondary exercises, performed as super-sets. The primary exercises were deadlift (trap bar, conventional or stiff-legged deadlift), squat (back or front squat), kettlebell swings, power clean, or Nordic Hamstring. These primary exercises were subject to variation between sessions, so that only three primary exercises were performed in each training session. Secondary exercises were low-load/body weight exercises for the upper body. The estimation of load in the primary exercises was individually calculated from a 3-repetition maximum (3 RM) test prior to the initiation of the intervention period [32]. After the intervention period, RT participants completed a POST-training 3 RM test. With the exception of Nordic Hamstring and kettlebell swings, all primary exercises were conducted as solely concentric movements. Subjects were instructed to use estimated loads as guidelines and to note the actual used load for the subsequent analysis of training progression. For this purpose, study participants were provided with a training logbook with individual estimations of training loads for the primary exercises. Subjects who completed the resistance training program showed a 91.0% self-reported adherence to training, with 19.5% of all sessions being supervised. CON participants followed regular off-season resistance training programs and were encouraged to conduct two sessions a week during the intervention period. The resistance training performed by CON did not include heavy-load (≥70% of 1 RM) slow or explosive-type exercises.

### 2.6. Statistical Analysis

Data were examined for a normal (Gaussian) distribution by visual inspection of Q-Q plots. A mixed linear model was used to evaluate interaction effects for PRE-to-POST changes for all variables, with the subject ID defined as a random effect and time and group as fixed effects. Paired Student’s *t*-testing was applied to test within-group differences of means between PRE- and POST-testing. The significance level was set at *p* ≤ 0.05 (two-tailed), and tendencies were defined as 0.05 < *p* ≤ 0.1, denoted as ^(^*^)^ for within-group and ^(#)^ for between-group differences. All statistical analyses were performed in Stata/BE 18.0 (StataCorp., College Station, TX, USA), and data are presented as means ± standard deviation (SD), while graphs were created using GraphPad Prism version 10.1.1 (GraphPad Software, Boston, MA, USA).

## 3. Results

### 3.1. Study Participants

A total of 27 players (RT *n* = 12, CON *n* = 15) completed the study (Table 1).

### 3.2. Countermovement Jump Performance

Jump height, calculated from toe-off (JH), improved by 1.5 cm (+4.8%, *p* = 0.012) and 2.3 cm (+8.4%, *p* = 0.044) in RT and CON, respectively, with no significant time-by-group interaction effect (*p* = 0.463) (Figure 4a and Table 3).

Jump height relative to the ground level (JH_GL_) improved by 3.2 cm for RT (+8.0%, *p* = 0.013), with no changes observed for CON (*p* = 0.185) and no time-by-group interaction effect (*p* = 0.158) (Figure 4b and Table 3). BCM_disp_ [Cp] showed a time-by-group interaction effect (*p* = 0.054), with RT increasing 3.1 cm (+7.8%, *p* = 0.032) after the training period and no within-group changes for CON (*p* = 0.916) (Figure 4c and Table 3). Data for produced work [Cp] revealed a within-group change of 0.4 joule/kg (+6.1%, *p* = 0.012) for RT after the training period, but no time-by-group interaction effect was observed (*p* = 0.220). A time-by-group interaction effect was demonstrated in peak F_z_ [Ep] (*p* = 0.005) with CON showing impaired peak force (−5.2%, *p* = 0.032) after the intervention period. No PRE to POST changes were observed for CMJ peak power, RFD 0–50 ms, peak F_z_ [Cp], mean F_z_ [Cp], BCM_disp_ [Ep] or T (Ep_dec_/Cp) (*p* > 0.05).

### 3.3. Knee Extensor/Flexor Strength and RTD

A non-significant time-by-group interaction effect (*p* = 0.055) for knee extensor MVIC peak torque was observed, with RT participants displaying a significant within-group POST-training change of 0.19 Nm/kg (+4.5% *p* = 0.044). (Figure 5 and Table 4).

No other PRE-to-POST changes were observed in mechanical muscle function (knee flexor peak torque, RTD) (Table 4).

### 3.4. Sprint Performance

In the 20-m sprint test, CON showed an impaired performance at POST compared to PRE (+0.06 s, *p* = 0.002) when recorded at the 20-m split time (Figure 6 and Table 5). A time-by-group interaction with a *p*-value of 0.061 demonstrated a tendency for the change in 20-m sprint time to differ between RT and CON, with RT maintaining 20-m sprint performance from PRE to POST (*p* = 0.485) (Table 5). Sprint times recorded at 5 and 10 m did not demonstrate any time-by-group interaction or within-group changes from PRE to POST (*p* > 0.05).

### 3.5. Agility Performance

A time-by-group interaction effect (*p* < 0.001) was observed in the agility test, with RT participants performing faster (−0.18 s, *p* = 0.008) and CON slower (+0.34 s, *p* = 0.012), respectively, after the training period (Figure 7 and Table 6).

Further, a time-by-group interaction effect was observed (*p* = 0.012) (Figure 7) when performing the agility test with handball dribbles, where CON demonstrated a diminished performance (+0.20 s, *p* = 0.017) when comparing PRE to POST values. No within-group PRE to POST changes were observed for RT in the agility test with ball dribbles (*p* = 0.421).

## 4. Discussion

The present study intended to evaluate the effect of off-season resistance training in elite female team handball players on lower limb mechanical muscle function and sports-specific physical performance.

The main finding of the present study was that eight weeks of undulating heavy-load resistance training led to an improved on-court agility performance by 3.5%. Furthermore, RT demonstrated a maintained sprint capacity, while CON showed a reduced sprint- and agility performance following the eight-week off-season intervention period. While the maximal vertical jump height performance improved in both groups, JH_GL_ was found to increase in RT only. The magnitude of upward displacement of BCM during the concentric take-off phase (BCM_disp_ [Cp]) increased in RT but not CON, to reach higher levels in RT than CON, indicating an increased release height for BCM as a result of the training performed in RT. In addition, knee extensor peak torque was improved for RT at POST-testing compared to PRE-testing, tending to be different from CON. The observed improvements in RT occurred despite of a reduced total training volume of ~70%.

### 4.1. Sports-Specific Performance

Team handball involves a wide range of explosive-type (rapid execution, high RFD) locomotion activities such as sprints, jumps, cuttings, sidesteps, and changes of direction [24]. To meet these complex demands, acceleration capacity, along with maximal movement speed and agility performance, has been identified to play a central role in the physical profile of team handball players [33]. The present study comprised concurrent measurements of acceleration capacity (5-m and 10-m split times) and maximum sprint speeds (20-m split time), along with assessments of on-court agility performance using an agility test developed specifically for team handball players.

The agility test implemented in the present study comprised several sections, with different elements of lateral and frontal body movements, with high demands for rapid muscle recruitment involving several interlinked short-distance (3-m to 5-m) accelerations performed at maximal voluntary effort (for more details, see Section 2). Although improved agility performance was observed POST-training in RT, it remains difficult to identify which element(s) (sections) of the test were responsible for the observed improvements. The agility test was initiated by a short (3-m) sprint acceleration, and likewise the last section of the test consisted of a short (5-m) sprint. Given that no changes were observed for RT with training in 5-m and 10-m acceleration capacity during the 20-m sprint test, it is plausible that the improved agility performance observed reflects an increased ability to perform rapid lateral sidecutting movements, changes of direction, and lateral side steps or improved (horizontal) jump capacity.

Proficiency in performing changes of directions has previously been emphasized as a critical skill within intermittent ball team sports [33,34] such as basketball [35] and football [36]. Change of direction is an SSC movement, demanding high levels of eccentric and concentric muscle strength and RFD. An improved output of a multitude of complex muscle actions would therefore be expected to be involved in any given training-induced enhancements in agility performance. In the present study, maximal knee extensor strength (peak torque) increased POST-training in RT, which could form the base for an improved agility performance in concentric propulsive phases. This is supported by an increase in SSC concentric work [Cp], indicating an enhanced ability to rapidly generate force.

The present resistance training regime did not improve sprint capacity in our cohort of elite female team handball players. Sprint capacity at PRE-testing for RT was comparable to other elite team sport athletes [37,38] and given the short time frame employed in the present study, it might take a longer time to improve sprint capacity with heavy-load resistance training. Notably however, the 20-m sprint performance was reduced in CON when assessed at the end of the eight-week intervention period, whereas no impairments were observed in RT, indicating that the stimuli of conventional off-season resistance training (as performed in CON) for elite female team handball players may not be effective in maintaining their 20-m sprint performance during the off-season period, in contrast to that observed with the undulated heavy resistance training program performed by RT participants. Along similar lines, agility performance with and without dribbles was impaired in CON after the intervention period, whereas significant improvements in the agility test performance were observed in RT.

### 4.2. Countermovement Jump Performance

The vast involvement of various and forceful SSC locomotion activities in team handball sets high demands for multi-joint muscle performance such as in sprinting, jumping, cutting, and other changes of direction. Consequently, a high transferability from CMJ testing to on-court performance may be expected. In support of this notion, the present RT participants were found to improve their vertical jump performance following the 8-week resistance training protocol, as manifested by increases in JH (+1.5%), propulsive work [Cp] (0.4%), BCM_disp_ [Cp] (+3.1%), and JH_GL_ (+3.2%). Previous studies investigating CMJ performance have reported significant improvements in maximal JH with resistance training in untrained [27] and trained [14,39] individuals, although different calculation methods for JH were employed, making direct comparisons difficult [26].

From a biomechanical perspective, BCM_disp_ [Cp] is an important variable to investigate, since the mechanical work produced by the leg extensor muscles during the concentric take-off phase is proportional to BCM_disp_ [Cp], because work = Force · Distance, where Force = mean vertical ground reaction force (mean F_z_) and Distance = BCM_disp_ [Cp]. The amount of work produced during the concentric take-off phase (work [Cp]) is responsible for generating the total mechanical energy (kinetic energy [E_kin_] + potential energy [E_pot_]) delivered to the body center of mass (BCM) at the instant of take-off, where E_kin_ = ½·BM·V_to_^2^ and E_pot_ = BM·g·BCM_disp_[Cp], where g = 9.81 m/s^2^. In turn, the vertical take-off velocity at toe-off (V_to_) determines the vertical jump flight height (JH, relative to toe-off) due to the conservation in mechanical energy during the flight phase: E_kin_ + E_pot_ = constant ⇒ JH = V_to_^2^/2 g. A longer work distance (BCM_disp_ [Cp]) is thus the main explanation for the increased work [Cp] observed after the training intervention for RT, since the mean F_z_ [Cp] remained unchanged from PRE- to POST-training. Furthermore, an impaired peak F_z_ [Ep] was observed in CON after the intervention period, different from RT, indicating a reduced peak deceleration force as a consequence of the off-season period.

Concentric work exerted on the BCM was found to increase (+6%) in RT, while remaining unaltered in CON (cf. Table 3). Further, since jump height relative to the ground level (JH_GL_) is dependent on BCM_disp_ [Ep], BCM_disp_ [Cp] and flight height (specifically JH_GL_ = JH + BCM_disp_[Cp] − BCM_disp_[Ep]), the observed increase in BCM_disp_ [Cp] was the main reason for the increased vertical jump height relative to the ground after the period of heavy-load resistance training.

The muscle groups involved in plantar flexion of the ancle joint are the gastrocnemius and soleus [40]. It may be suggested that RT participants performed a more extended range of motion during the plantar flexion movement in the final concentric take-off phase. This could increase both BCM_disp_ [Cp] and the magnitude of mechanical work produced on BCM during the concentric take-off phase, thereby leading to an increased jump height above the ground level.

### 4.3. Mechanical Muscle Function

The resistance training-induced improvements in knee extensor MVIC peak torque may be explained by increased neural drive (i.e., increased MU firing frequencies and/or or more full motor unit recruitment) and gains in muscle fiber size (hypertrophy) [5]. Previous reports from our lab have demonstrated elevated EMG signal amplitudes in the knee extensor muscles after 16 weeks of resistance training in men [5], founding the base for an increased neural drive after training. In the present study, EMG recording was not performed. Future research should consider including EMG measurements in the knee extensors and flexors to elucidate the potential effects of (short-term) resistance training on neural drive in well-trained female elite athletes.

The lack of improvements in knee flexor MVIC torque and RTD for both the knee extensors and flexors, respectively, may be related to somewhat limited duration (8 wk) and a low total workload (16 sessions). Thus, RT performed two training sessions per week, with one weekly session focusing on explosive muscle actions and the other session intending to stimulate muscle hypertrophy and fatigue resistance by the accumulation of metabolic stress. In result, eight designated sessions were performed for each sub-target (RFD and hypertrophy/fatigue resistance), which may have been insufficient for inducing gains in RFD. In the present study, coupled concentric–eccentric movements with free weights involving multi-joint exercises were employed during all training sessions. This approach to resistance training deviates from prior studies that have reported an enhanced rate of force capacity through progressive heavy resistance training. Those studies utilized controlled exercises in resistance training machines [5,6,7], potentially introducing a confounding factor that may account for the absence of enhanced RTD and knee flexor strength observed in the present study.

### 4.4. Limitations

A number of limitations may be mentioned with the present study. Firstly, 45% of the included participant completed the study. Three subjects were excluded from the study during the intervention period due to meeting one or more of the exclusion criteria. The rest of the subjects (n = 33) withdraw due to external factors, not directly related to the study. The main reason for dropouts was logistical obstacles, indicating that a substantial proportion of the players were challenged by difficulties in balancing work/school with a professional team handball career in this off-season phase. Further, a number of coaches and physical trainers in the two control clubs unexpectedly left their positions during the initial phase of the observation period, which hindered communication with the involved players and may have caused inconsistent protocols of off-season resistance training in CON participants. Formally, CON participants were encouraged to perform their regular program of off-season resistance training twice a week, comprising exercise loads not exceeding 70% of 1 RM (maintenance workout). Future studies should consider the employment of more rigorously controlled resistance training programs.

### 4.5. Practical Implications

The designated off-season resistance training program investigated in the present study may be implemented by coaches and practitioners to maintain and optimize physiological capacity during off-season periods in elite female team handball players, using a low-frequency training protocol involving two sessions per week. Furthermore, the present exercise protocol may hold potential for off-season resistance training in athletes engaged in other types of high-intensity intermittent sports, characterized by the strong involvement of maximal SSC muscle actions such as in maximal vertical jumping and change-of-direction movements (i.e., football, volleyball, basketball, badminton, tennis, squash, rugby, and ice hockey).

## 5. Conclusions

As a key finding of the present study, team handball-specific agility performance was improved in response to an eight-week undulated heavy-load resistance training program performed during the off-season period in female elite team handball players, representing an important functional gain in this athlete population. Furthermore, signs of improved SSC capacity, accompanied by increases in maximal knee extensor strength (MVIC peak torque), were demonstrated in RT after the training period. The improved CMJ JH performance observed in RT could be explained by increased mechanical work production during the concentric take-off phase, potentially reflecting the enhanced capacity of the hip and knee extensors, as well as the plantar flexors, to contribute to an enhancement in maximal vertical jump height. An increased knee extensor MVIC peak torque suggests the involvement of neural and/or muscular (hypertrophic) adaptations. In contrast, no changes were observed in maximal knee flexor strength (MVIC peak torque) or in any measures of explosive muscle strength (RTD). The present data collectively suggest that undulating heavy-load resistance training during the off-season is effective in not only maintaining but improving sports-specific performance (handball-specific agility, sprint capacity, vertical jump height) and maximal knee extensor muscle strength in elite female team handball players.

## Figures and Tables

**Figure 1 jfmk-09-00268-f001:**
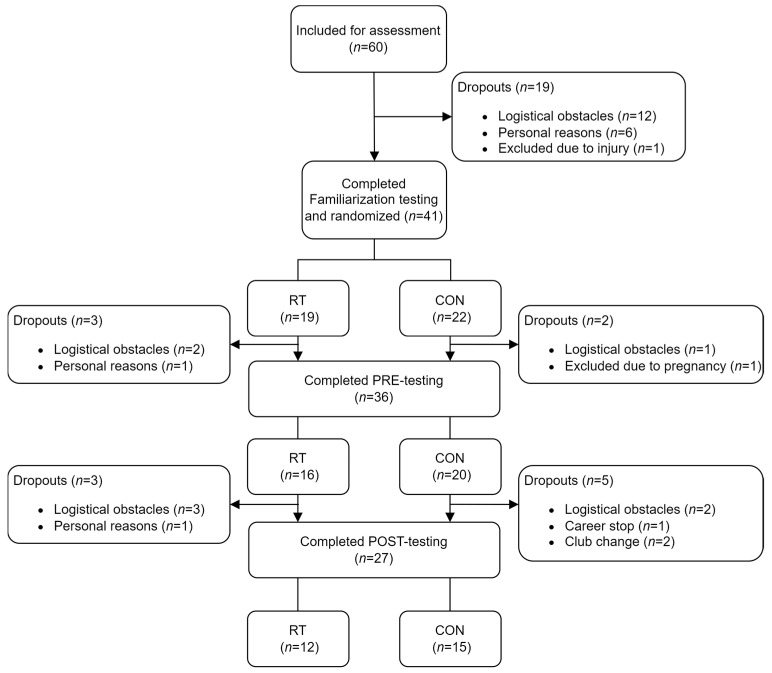
Flow chart of study participants. Figure presents number of study participants completing the three test sessions (Familiarization, PRE, POST), divided (randomized) into the resistance training group (RT) and the training-as-usual control group (CON). The number of dropouts and the corresponding reason for dropping out/exclusion are also presented.

**Figure 2 jfmk-09-00268-f002:**
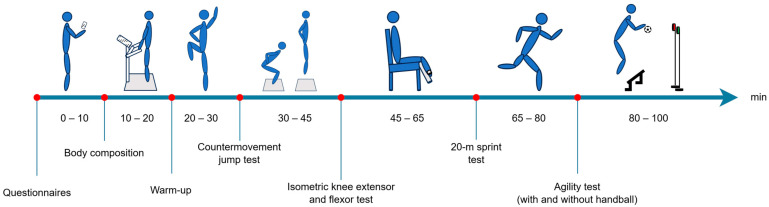
On-field test protocol. Timeline and order of evaluated tests. Study participants filled out an online questionnaire before having their body composition measured. Warm-up procedures were completed before engaging in countermovement jump test, isometric knee extensor and flexor test, 20 m sprint test, and agility test with and without handball.

**Figure 3 jfmk-09-00268-f003:**
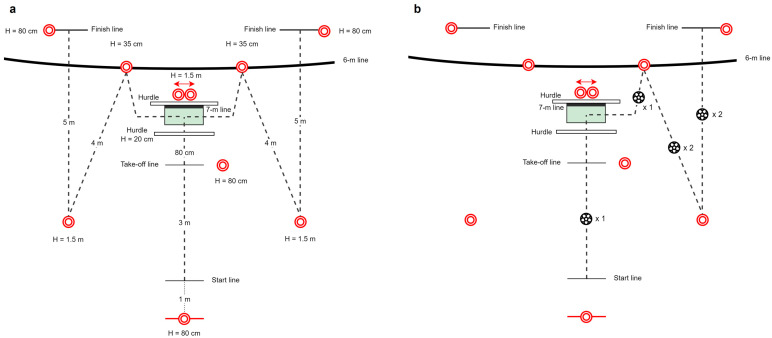
Agility test. Schematic presentation of the agility test (**a**), delineating the trajectory of players through dotted lines. The bold dark line is fixed markings on the handball court (6-m and 7-m lines, respectively). Red dots indicate the positioning of touch and motion sensors. The agility test performed with a handball (**b**), delineating the specific sections of the test wherein dribbling maneuvers are to be executed.

**Figure 4 jfmk-09-00268-f004:**
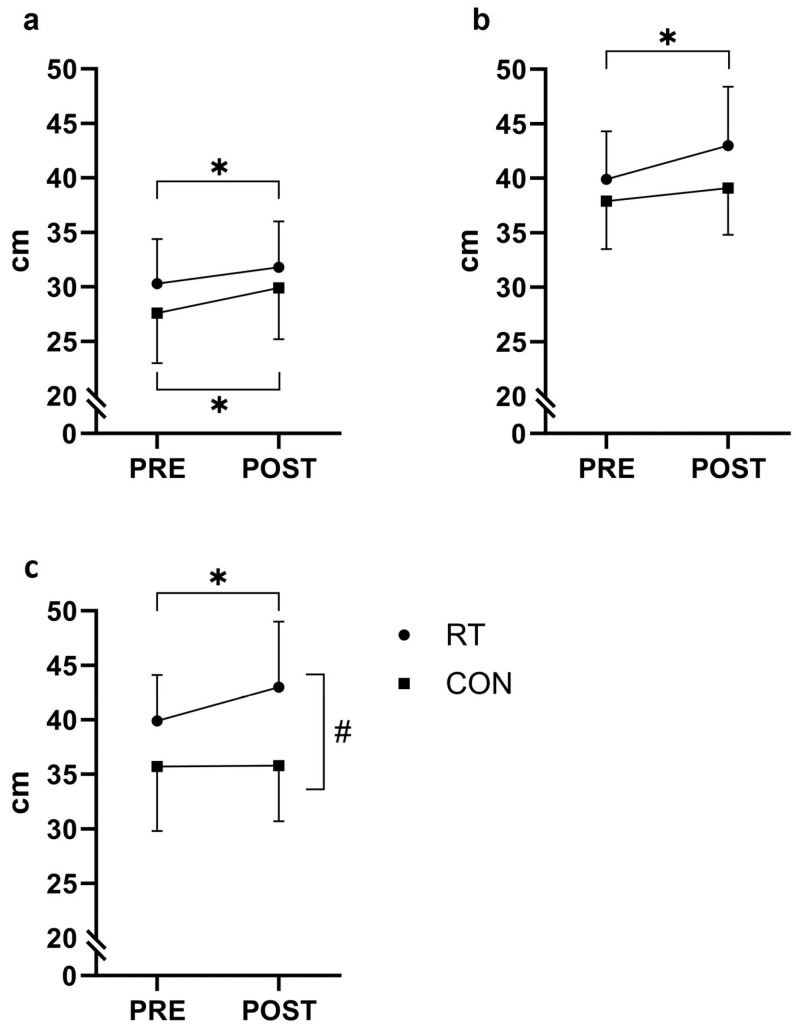
Countermovement jump. (**a**) Jump height (JH), (**b**) jump height relative to ground level (JHGL), (**c**) body center of mass displacement in the concentric (upwards) phase (BCMdisp [Cp]). Mean ± standard deviation values before (PRE) and after (POST) the eight-week intervention period for the resistance training group (RT) and the control group (CON). ^#^
*p* ≤ 0.05 time-by-group interaction effect, * *p* ≤ 0.05 within-group change.

**Figure 5 jfmk-09-00268-f005:**
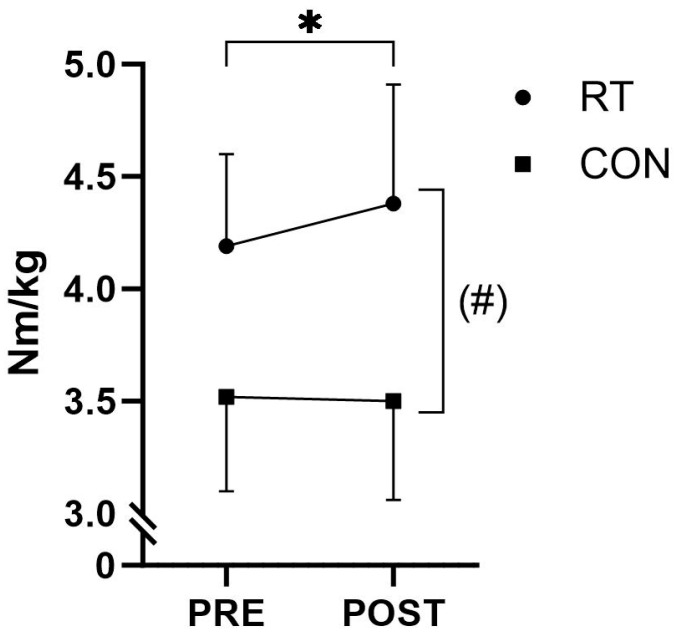
Knee extensor MVIC peak torque. Knee extensor peak torque (group means ± standard deviation) before (PRE) and after (POST) the intervention period for the resistance training group (RT) and the control group (CON). ^(#)^ *p* ≤ 0.10 time-by-group interaction effect, * *p* ≤ 0.05 within-group change.

**Figure 6 jfmk-09-00268-f006:**
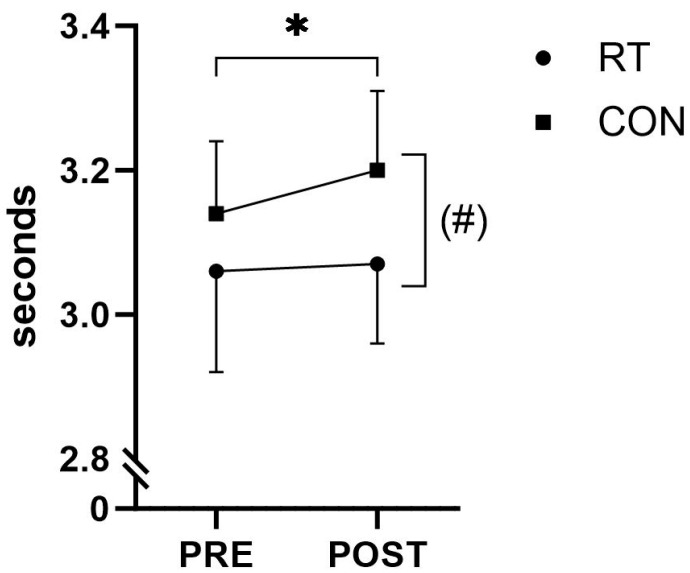
Twenty meter sprint performance. Twenty meter sprint performance (group means ± standard deviation) before (PRE) and after (POST) the intervention period for the resistance training group (RT) and the control group (CON). ^(#)^ *p* ≤ 0.10 time-by-group interaction effect, * *p* ≤ 0.05 within-group change.

**Figure 7 jfmk-09-00268-f007:**
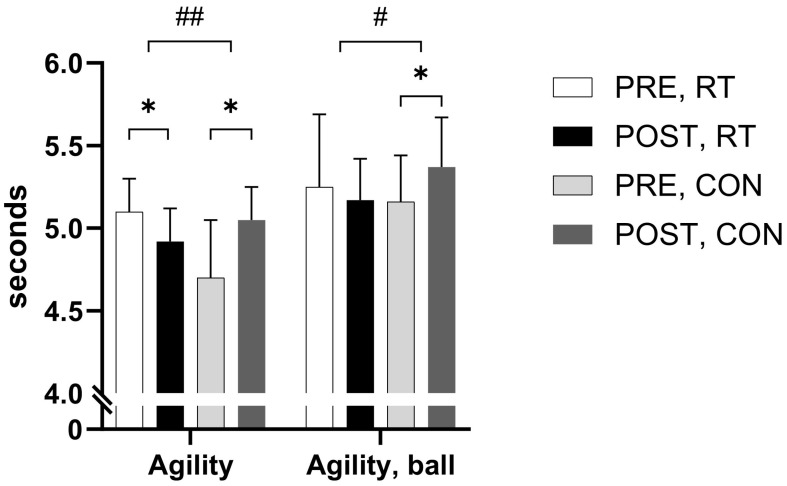
Agility performance. Figure shows means ± standard deviation of time to complete the change of direction test without (agility) and with ball (agility, ball) before (PRE) and after (POST) the intervention period for the resistance training group (RT) and the control group (CON). * *p* ≤ 0.05 within-group change, ^#^ *p* ≤ 0.05 time-by-group interaction effect, ^##^ *p* ≤ 0.00 time-by-group interaction effect.

**Table 1 jfmk-09-00268-t001:** Characteristics. Characteristics of study participants in the resistance training (RT) and control (CON) group before (PRE) and after (POST) the eight-week intervention period.

	RT *n* = 12		CON *n* = 15	
	PRE Mean ± SD	POST Mean ± SD	PRE Mean ± SD	POST Mean ± SD
Age (years)	23.0 ± 2.7	23.3 ± 2.6	24.1 ± 3.9	24.3 ± 3.8
Height (cm)	175.5 ± 4.1	176.4 ± 4.2	176.3 ± 4.9	176.2 ± 4.9
Body mass (kg)	72.9 ± 7.1	72.9 ± 7.5	74.9 ± 7.4	75.5 ± 8.0
Skeletal muscle mass (kg)	35.5 ± 3.1	33.7 ± 2.9	32.9 ± 2.7	32.8 ± 2.8
Body fat percentage (%)	18.5 ± 4.0	17.9 ± 4.2	21.5 ± 5.2	22.4 ± 5.2
Years of playing team handball (years)		15.7 ± 4.4		17.4 ± 4.3
Years of playing on elite level (years)		6.4 ± 2.6		7.9 ± 3.8
Team handball-related activities per week (*n*)		6.9 ± 0.7		8.0 ± 1.2
Hours of team handball-related training activities per week (h)		11.9 ± 2.4		13.2 ± 3.4
Experience with resistance training (years)		8.1 ± 2.0		10.1 ± 3.1
Playing position (*n*):				
Goalkeepers		2		2
Center backs (playmakers)		1		3
Backs (left/right)		5		4
Wings (left/right)		2		6
Center forwards (pivots)		2		0

**Table 2 jfmk-09-00268-t002:** Resistance training program. Overview of the eight-week program, with primary and secondary exercises, sets, reps, and estimated load for the explosive (Session #1) and slow (Session #2) sessions, respectively.

	Weekly Session #1—EXPLOSIVE Execution (Maximal Intentional Speed)							Weekly Session #2—SLOW Execution (Long Time Under Tension)						
Week	Primary Exercise	Variables			Secondary Exercise	Variables		Primary Exercise	Variables			Secondary Exercise	Variables	
		Sets	Reps.	Load (RM)		Reps.	Load (kg)		Sets	Reps.	Load (RM)		Reps.	Load (kg)
1	Trap bar deadlift	3	10	12	Push ups	10–12	-	Conventional deadlift	4	10	10	Lying leg raises	10	-
	Back squat	3	10	12	Pull ups	8–10	-	Back squat	4	10	10	Unilateral dumbbell bent over bench row	10 × 2	Light (10–20)
	Kettlebell swings	3	10	12	Weighted sit ups	10	10	Nordic hamstring	4	10	10	Rotator cuff exercise with powerband	10 × 2	-
2	Trap bar deadlift	3	8	10	Unilateral ankle mobility in powerband	8 × 2	-	Romanian deadlift	4	12	12	Plank with lateral rotations	6 × 2	-
	Back squat	3	8	10	Overhead press	10–12	Light (20–30)	Front squat	4	12	12	Weighted sit ups	10	10
	Power clean	3	8	10	Russian twist	20	10	Kettlebell swings	4	12	12	Hamstring mobility	-	-
3	Romanian deadlift	3	6	8	Lying leg raises	10	-	Trap bar deadlift	4	8	8	Lateral side steps with powerband	8 × 2	-
	Back squat	3	6	8	Pull down	10	Light (20–30)	Box squat (touch & go)	4	8	8	Side plank with leg adduction	8 × 2	-
	Kettlebell swings	3	6	8	Unilateral jump to bosu ball	6 × 2	-	Nordic hamstring	4	8	8	Push press	8	Moderate (30–50)
4	Trap bar deadlift	4	6	8	Weighted sit ups	10	10	Conventional deadlift	4	10	10	Unilateral ankle mobility in powerband	10 × 2	-
	Back squat	4	6	8	Copenhagen hip adduction	6 × 2	-	Back squat	4	10	10	Russian twist	20	10
	Kettlebell swings	4	6	8	Chin ups	6–8	-	Nordic hamstring	4	10	10	Push ups	10–12	-
5	Trap bar deadlift	3	4	6	Plank with lateral rotations	4 × 2	-	Romanian deadlift	4	8	8	Unilateral powerband pull down with abdominal rotation	8 × 2	-
	Back squat	3	4	6	Push ups	10–12	-	Front squat	4	8	8	Ab wheel roll outs	8 × 2	-
	Power clean	3	4	6	Unilateral ankle mobility in powerband	8 × 2	-	Kettlebell swings	4	8	8	Hamstring mobility	-	-
6	Romanian deadlift	3	4	6	Overhead press	10–12	Light (20–30)	Trap bar deadlift	4	12	12	Unilateral dumbbell bent over bench row	12 × 2	Light (10–20)
	Back squat	3	4	6	Lying leg raises	10	-	Box squat (touch & go)	4	12	12	Unilateral ankle mobility in powerband	12 × 2	-
	Kettlebell swings	3	4	6	Unilateral jump to bosu ball	4 × 2	-	Nordic hamstring	4	12	12	Chin ups	10–12	-
7	Trap bar deadlift	4	6	8	Plank with lateral rotations	6 × 2	-	Conventional deadlift	4	10	10	Copenhagen hip adduction	10 × 2	-
	Back squat	4	6	8	Pull ups	10–12	-	Back squat	4	10	10	Weighted sit ups	10	10
	Kettlebell swings	4	6	8	Russian twist	20	10	Nordic hamstring	4	10	10	Unilateral powerband pull up with abdominal rotation	10 × 2	-
8	Trap bar deadlift	3	4	6	Unilateral rotator cuff exercise, powerband	10 × 2	-	Romanian deadlift	4	8	8	Push ups	10–12	-
	Back squat	3	4	6	Overhead press	10–12	Light (20–30)	Front squat	4	8	8	Unilateral dumbbell bent over bench row	8 × 2	Light (10–20)
	Power clean	3	4	6	Abdominal wheel roll outs	10–12	-	Kettlebell swings	4	8	8	Side plank with leg adduction	8 × 2	-

**Table 3 jfmk-09-00268-t003:** Countermovement jump kinematics and kinetics. Variables assessed from the countermovement jump testing (group means ± standard deviation (SD)) of the resistance training group (RT) and control group (CON), before (PRE) and after (POST) the eight-week intervention period. JH_GL_ = jump height relative to ground level, [Cp] = concentric phase, [Ep] = eccentric phase, [Ep_dec_] = eccentric deceleration phase, RFD = rate of force development, F_z_ = vertical ground reaction force, BCM_disp_ = body center of mass displacement, T = time. Values are displayed as absolute change (Abs. change), percentage change (% change), within-group *p*-value (* *p* ≤ 0.05, ^(^*^)^ 0.05 < *p* ≤ 0.1), and time-by-group interaction effects *p*-value (^#^ *p* ≤ 0.05).

	RT *n* = 12					CON *n* = 14					
	PRE Mean ± SD	POST Mean ± SD	Abs. Change	%-Change	Within-Group *p*-Value	PRE Mean ± SD	POST Mean ± SD	Abs. Change	%-Change	Within-Group *p*-Value	Interaction Effects (Time × Group) *p*-Value
Jump height cm	30.3 ± 4.1	31.8 ± 4.2	1.5 *	4.8	0.012	27.6 ± 4.6	29.9 ± 4.7	2.3*	8.4	0.044	0.463
JH_GL_ cm	39.9 ± 4.4	43.0 ± 5.4	3.2 *	8.0	0.013	37.9 ± 4.4	39.1 ± 4.3	1.3	3.4	0.185	0.158
Peak power [Cp] Watt/kg	45.3 ± 4.9	45.6 ± 5.4	0.3	0.7	0.627	44.5 ± 4.7	44.1 ± 4.4	−0.4	−0.9	0.631	0.483
RFD 0–50 ms [Ep_dec_] N·s^−1^/kg	111.6 ± 53.5	113.6 ± 41.9	1.9	1.7	0.867	144.6 ± 50.9	118.9 ± 39.9	−25.7	−17.8	0.109	0.136
Work [Cp] Joule/kg	6.84 ± 0.67	7.26 ± 0.78	0.42 *	6.1	0.012	6.19 ± 0.77	6.39 ± 0.86	0.20	3.3	0.117	0.220
Peak F_z_ [Ep] N/kg	21.9 ± 2.1	22.6 ± 1.4	0.7	3.1	0.190	23.3 ± 2.1	22.1 ± 1.9	−1.2 *	−5.2	0.032	0.005 ^#^
Peak F_z_ [Cp] N/kg	20.4 ± 5.3	20.6 ± 5.3	0.2	1.1	0.928	17.8 ± 7.5	21.9 ± 3.9	4.0	22.5	0.126	0.209
Mean F_z_ [Cp] N/kg	18.7 ± 1.4	18.6 ± 1.5	−0.1	−0.6	0.736	19.5	19.4	−0.1	−0.3	0.914	0.872
BCM_disp_ [Ep] cm	30.3 ± 3.7	31.7 ± 5.4	1.4	4.6	0.329	25.5 ± 4.9	26.6 ± 5.8	1.1	4.5	0.186	0.868
BCM_disp_ [Cp] cm	39.9 ± 4.2	43.0 ± 6.0	3.1 *	7.8	0.032	35.7 ± 5.9	35.8 ± 5.1	0.1	0.3	0.916	0.054 ^#^
T [Ep_dec_] ms	168.3 ± 30.1	165.3 ± 26.6	−3.1	−1.8	0.695	136.4 ± 25.1	146.8 ± 31.8	10.4 ^(^*^)^	7.6	0.092	0.136
T [Cp] ms	273.3 ± 31.1	282.8 ± 37.6	9.6	3.5	0.327	245.1 ± 40.3	250.5 ± 34.0	5.4	2.2	0.459	0.702

**Table 4 jfmk-09-00268-t004:** Knee extensor and flexor MVIC peak torque and rate of torque development. Knee extension and knee flexion peak torque and rate of torque development (RTD), normalized to body mass, of the resistance training group (RT) and control group (CON) before (PRE) and after (POST) the eight-week intervention period. Values are displayed as group means ± standard deviation (SD), absolute change (Abs. change), percentage change (%-change) within-group *p*-value (* *p* ≤ 0.05) and time-by-group interaction effects *p*-value (^(#)^ 0.05 < *p* ≤ 0.1).

		RT *n* = 12					CON *n* = 12					
		PRE Mean ± SD	POST Mean ± SD	Abs. Change	%-Change	Within-Group *p*-Value	PRE Mean ± SD	POST Mean ± SD	Abs. Change	%-Change	Within-Group *p*-Value	Interaction Effects (Time × Group) *p*-Value
Knee extensor	MVIC peak torque Nm/kg	4.19 ± 0.41	4.38 ± 0.53	0.19 *	4.5	0.044	3.52 ± 0.42	3.50 ± 0.44	−0.02	−0.6	0.805	0.055 ^(#)^
	RTD 0–30 ms Nm·s^−1^/kg	21.1 ± 5.8	21.0 ± 6.3	−0.1	−0.4	0.961	17.9 ± 8.0	16.1 ± 9.8	−1.8	−10.2	0.208	0.435
	RTD 0–100 ms Nm·s^−1^/kg	22.4 ± 2.3	21.6 ± 3.1	−0.8	−3.7	0.163	20.0 ± 5.0	18.6 ± 5.6	−1.4	−6.8	0.086	0.534
Knee flexor	MVIC peak torque Nm/kg	1.91 ± 0.25	1.94 ± 0.21	0.03	1.7	0.468	1.73 ± 0.15	1.73 ± 0.13	0.00	−0.17	0.932	0.502
	RTD 0–30 ms Nm·s^−1^/kg	4.0 ± 1.4	3.4 ± 1.5	−0.6	−14.5	0.375	3.7 ± 1.8	3.4 ± 1.9	−0.4	−10.0	0.354	0.774
	RTD 0–100 ms Nm·s^−1^/kg	8.3 ± 2.1	8.0 ± 1.9	−0.3	−3.2	0.744	8.3 ± 2.6	8.2 ± 2.3	−0.1	−0.7	0.872	0.804

**Table 5 jfmk-09-00268-t005:** Sprint performance (group means ± standard deviation (SD)) assessed in a 20-m sprint test, with split times at 5-m, 10-m, and 20-m, for the resistance training group (RT) and control group (CON), before (PRE) and after (POST) the eight-week intervention period. Values are displayed as absolute change (Abs. change), percentage change (%-change), within-group *p*-value (* *p* ≤ 0.05, ^(^*^)^ 0.05 < *p* ≤ 0.1), and time-by-group interaction effects *p*-value (^(#)^ 0.05 < *p* ≤ 0.1).

	RT *n* = 12					CON *n* = 14					
	PRE Mean ± SD	POST Mean ± SD	Abs. Change	%-Change	Within-Group *p*-Value	PRE Mean ± SD	POST Mean ± SD	Abs. Change	%-Change	Within-Group *p*-Value	Interaction Effects (Time × Group) *p*-Value
5-m seconds	0.87 ± 0.04	0.89 ± 0.03	0.02 ^(^*^)^	1.7	0.098	0.94 ± 0.03	0.93 ± 0.06	−0.01	−1.1	0.449	0.110
10-m seconds	1.66 ± 0.08	1.67 ± 0.07	0.01	0.7	0.300	1.74 ± 0.07	1.76 ± 0.07	0.03	1.4	0.180	0.487
20-m seconds	3.06 ± 0.14	3.07 ± 0.11	0.01	0.5	0.485	3.14 ± 0.10	3.20 ± 0.11	0.06	1.9 *	0.002	0.061 ^(#)^

**Table 6 jfmk-09-00268-t006:** Agility performance. Agility test performance (group means ± SD) when assessed with (agility, ball) or without handball (agility) in the resistance training group (RT) and control group (CON), before (PRE) and after (POST) the eight-week intervention period. Values are displayed as absolute changes (Abs. change), percentage changes (% change), within-group *p*-value (* *p* ≤ 0.05), and time-by-group interaction effects *p*-value (^#^ *p* ≤ 0.05).

	RT *n* = 12					CON *n* = 11					
	PRE Mean ± SD	POST Mean ± SD	Abs. Change	%-Change	Within-Group *p*-Value	PRE Mean ± SD	POST Mean ± SD	Abs. Change	%-Change	Within-Group *p*-Value	Interaction Effects (Time × Group) *p*-Value
Agility seconds	5.10 ± 0.20	4.92 ± 0.20	−0.18 *	−3.5	0.008	4.70 ± 0.35	5.05 ± 0.20	0.34 *	7.3	0.012	<0.001 ^#^
Agility, ball seconds	5.25 ± 0.44	5.17 ± 0.25	−0.07	−1.4	0.421	5.16 ± 0.28	5.37 ± 0.30	0.20 *	4.0	0.017	0.012 ^#^

## Data Availability

Data is contained within the article.

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
