# Peer review of "Effects of Off-Season Heavy-Load Resistance Training on Lower Limb Mechanical Muscle Function and Physical Performance in Elite Female Team Handball Players"

_jfmk, 2024, doi:10.3390/jfmk9040268_

Round 1

Reviewer 1 Report

Comments and Suggestions for Authors

Review - Effects of off-season heavy-load resistance training on lower  limb mechanical muscle function, and sports specific performance in elite female team handball players

General  comments –  

Since the authors have selected an interesting research topic, their work should be acknowledged. However, the structure of the manuscript should be revised by the authors. There is a large disequilibirum between the introduction, methods and discussion sections. My reading follows the variables studied in this paper. However, in none point of the introduction section the authors present the state of the art of those variables in the studies conducted about mechanical function in elite athletes. Going to the methods sections, the authors show an amount of variables that few of them were presented in the introduction section. The same problem is around the discussion section, when authors were too superficial based on the variables presented on the methods and the results sections. With this observation, the meaning of the study is compromised. I suggest the authors to carry out a more detailed bibliographic review to go deeper in some relevant variables presented in this study, including in the introduction and discussion section. With these adjustments, the  research question and the meaning of this paper will be more clear.

There are also some minor comments that I detected in this present paper, and I hope that can help the authors. 

Specific comments –

Title

 delete the words  “and sports specific performance”

Abstract

L 21 – is well known

L23 – Please, include this sentence and delete the previous one : The current study aimed to analyze the effects of...

L26 to L27 – Include a brief sentence about the experimental protocol (tests, instruments, variables collected)

L38 to L39 – Include a brief sentence about the practical applications of this study.

Introduction

In this section, there is a disequilibrium among the paragraphs. There is paragraphs with 4 lines and others with 20. Please rewrite them, establishing an equilibrium among them. Another point, is that the research question is not clear in this study.  In this study also, there is no hypothesis, why? If the auhtors quoted the Fristrup study´s to raise the objective of this research. Another point, is that the authors in none moment related the aspect to evaluate the handball athletes out of the season? What are the implications for training?

L50 to L51 – There is no connection between these paragraphs. In the first para, authors describe the detraining condition in female handball players. In the second para, the authors started describing the characteristics of the female handball modality... does not make sense. 

L69 – This para does not make sense here. Upload this information to the previous paragraph.

Methods

L100 to L111 – Draw a flow chart for a better explanation of the handball athlete recruitment.

L125 to L131 -  This paragraph is not well explained. Please, describe how the tests were conducted. The intervals between them, number of repetitions...

L133 to 139 – This information described here are not in accordance with the sequence of the text.

L173 to L174 – The authors are encouraged to insert a figure from matlab, explaining the phases of the movements and the kinetic variables assessed.

L206 to L208 – Quote a pair of references  that describe the submaximal and maximal muscle contraction protocol.

L190 to L232- How the kinetic data were normalized?

L242 to L244 – How authors determined 60 and 80% intensity?

L256 to L262 – It will be suitable to patronize the unit description. Ex., 4-m or 4m, 1.5-m or 1.5m

L282 to L295 – How authors adopted the load of exercise? Heavy load...lower body exercise?  

Figure 3 – My suggestion is to insert the intervals between the repetitions of each exercise proposed.

Results

L328 to L332 – Why this informtion is here?  Please realocate for method section.

Table 2 and Table 3 – The description of the variables included in these tables  should be as a note.

Discussion

Based on the results for each variable presented in this study, the discussion is a little bit poor according to the results presented. The authors are too superficial based on the amount of the results presented.  Also, in this section, the authors should include a para  describing the discussion about the methodology adopted. 

L403 – A overall discussion about these results should be included.

Reviewer 2 Report

Comments and Suggestions for Authors

The article is well-structured methodologically, with its greatest strength in the description of the methods section. However, I modestly believe that a physical intervention during the transition period (off-season) inevitably has significant effects on physical performance. Since the study does not use independent samples, it cannot reliably determine, through scientific methodology, that the intervention has notable implications compared to other types of interventions, nor that the effects of the implemented physical stimulus have direct impacts on performance, as significant external variables were not controlled.

Nevertheless, the following recommendations are suggested for future research:

  1. Compare the intervention process used in the study with other training alternatives already implemented by other authors.
  2. Ensure the study includes a representative sample of the population. Otherwise, use G*Power to determine the necessary sample size, statistical power, and effect size.
  3. Consider redesigning the title of the article based on the results achieved. The title specifies "Specific Sports Performance," but the type of performance is unclear until the full document is read (the author describes agility, sprint, and jump).
  4. In the first paragraph of the discussion section, describe the research objective and whether it was achieved.
  5. The limitations of the study should correspond to the research type, with attention to the non-representative sample that limits the generalizability of results.

I strongly believe that the research could be restructured as a quasi-experimental study, provided it includes a control group for comparison. Under this framework, I recommend substantially modifying the research and re-evaluating it.

Reviewer 3 Report

Comments and Suggestions for Authors

Many thanks to the authors for their contribution to the current field of research. I think that is a well-written, but some minor comments may be needed. 

Below, you can find my minor comments. 

1. Please include the power of your study according to your sample size in the statistical analysis paragraph. 

In general, it is a vary constructive manuscript for the readers. I belive that it can proceed for publications as it is. 

Reviewer 4 Report

Comments and Suggestions for Authors

The interesting idea of ​​this study, my recommendations are the following:

Abstract - recommend to mention the average age and the standard deviation of the subjects from the two groups.

Keywords - I recommend that they be mentioned according to the editing rules.

In the Methods section, I recommend the introduction of a new section called Study design where the typology of the study and the specific aspects can be presented.

I recommend that in the Subjects section the power of the sample be calculated by applying the statistical index G Power.

I recommend that lines 125-131 be moved from the Subjects section to the Instruments subsection.

I recommend that the subsections related to the Methods section be numbered.

Lines 151-309 recommend to be grouped in a new subsection called Instruments and Procedures. And the aspects concerned should be numbered according to this new subsection.

Fig 4 I recommend moving to the Subjects section from the Methods section. This chart is not a feature of the Results section.

Line 324 I recommend to mention subsections not subtitles.

Table 1 recommends clarifications regarding the height differences reported between the two tests, by group. The study was carried out over 8 weeks, and the subjects are adults.

According to the tables in the Results section, I recommend mentioning the differences in the number of subjects from the CON group, compared to the selection = 15 subjects.

Table 5 has no title, I recommend clarification.

At the end of the Discussions section, I recommend mentioning future research directions.

Discussions section - in subsections Countermovement jump performance - there is no correlation with information from previous studies.

Lines 409-413 recommend deletion, applied tests were described previously.

I recommend the expansion of the Discussion section by making concrete corrections between the results of this study and the results of previous studies regarding the monitored parameters.

Mechanical muscle function - bibliographic index 7 is mentioned throughout this subsection, I recommend expanding and mentioning other relevant bibliographic indexes.

According to the bibliography, the year 2022 is the last year of reference, in the period 2022-2024 no bibliographic source is mentioned. I recommend expanding and introducing sources from the last two years in the related sections.

Conclusions - the mention of sources 8-10 is not relevant, I recommend deleting them.

Round 2

Reviewer 2 Report

Comments and Suggestions for Authors

Considering the objective and level of the journal, as well as the responses provided by the authors of the research, I believe the manuscript has addressed most of the concerns raised in my evaluation.

Reviewer 4 Report

Comments and Suggestions for Authors

No comments